**Data Availability Statement:** All relevant data are within the manuscript and its Supporting Information files.

# Treatment outcome of Severe Acute Malnutrition and associated factors among under-five children in outpatient therapeutics unit in Gubalafto Wereda, North Wollo Zone, Ethiopia, 2019

**Biruk Beletew Abate**[ID]*, **Befkad Deresse Tilahun**[☯], **Ayelign Mengesha Kassie**[☯], **Mesfin Wudu Kassaw**[☯]

Department of Nursing, College of Health Sciences, Woldia University, Woldia, Amhara Regional State, Ethiopia

☯ These authors contributed equally to this work.
* birukkelemb@gmail.com

## Abstract

### Background

In Ethiopia, uncomplicated severe acute malnutrition is managed through the outpatient therapeutic program at health posts level. This brings the services for the management of Severe Acute Malnutrition closer to the community by making services available at decentralized treatment points within the primary health care settings. So far, evidence of the treatment outcome of the program is limited.

### Objective

The main aim of this study was to determine the magnitude of treatment outcomes of severe acute malnutrition and associated factors among under-five children at outpatient therapeutic feeding units in Gubalafto Wereda, Ethiopia, 2019.

### Methods

This was a retrospective cohort study conducted on 600 children who had been managed for Severe Acute Malnutrition (SAM) under Outpatient Therapeutic Program (OTP) in Gubalafto Wereda from April to May/2019. The children were selected using systematic random sampling from 9 health posts. The structured, pre-tested, and adapted questionnaire was used to collect the data. The data was entered by using EPI-data Version 4.2 and exported to SPSS version 24.0 for analysis. Bivariate and Multivariate regression was also carried out to determine the association between dependent and independent variables.

### Results

A total of 600 records of children with a diagnosis of severe acute malnutrition were reviewed. Of these cases of malnutrition, the recovery rate was found to be 65%. The death

**Funding:** The study was funded by Woldia University. However, the funder had no role in study design, data collection and analysis, decision to publish, or preparation of the manuscript.

**Competing interests:** The authors declare that they have no competing interests.

**Abbreviations:** EDHS, Ethiopian Demographic Health Serve; IMCI, Integrated Management of Childhood; MOH, Ministry of health; MUAC, Mid Upper Arm Circumference; MAM, Moderate Acute Malnutrition; OTP, Outpatient Therapeutic Program; RUTF, Ready to Use Therapeutic Food; SAM, Severe Acute Malnutrition; TFU, Therapeutic feeding unit; UNICEF, United Nations Children's Fund; WFH, Weight For Height.

rate, default rate, and medical transfer were 2.0, 16.0, and 17.0 respectively. Immunized children had 6.85 times higher odds of recovery than children who were not immunized (AOR = 6.85 at 95% CI (3.68–12.76)). The likelihood of recovery was 3.78 times higher among children with new admission than those with re-admission (AOR = 3.78at 95% CI ((1.77–8.07))). Likewise, children provided with amoxicillin were 3.38 times recovered than their counterparts (AOR = 3.38 at 95% CI ((1.61–7.08))). SAM treatment in OTP is beneficial because of its local access for most severe cases since children reach early before developing complications as a result fatalities will be reduced.

## Conclusions

The recovery rate and medical transfer were lower than the sphere standard. Presence of cough, presence of diarrhea admission category, provision of amoxicillin, and immunization status were factors identified as significantly associated with treatment outcome of severe acute malnutrition. The impact on increasing the recovery rates of children treated using the OTP service indicates the potential benefits of increasing the capacity of such services across a target region on child mortality/recovery. Timely intervention is another benefit of a more local service like OTP. Building capacity of OTP service providers and regular monitoring of service provision based on the management protocol was recommended.

## Introduction

Severe Acute Malnutrition (SAM) is defined by very low weight for height (below -3z scores of the median WHO growth standards), by visible severe wasting, or by the presence of oedema of both feet and mid-upper arm circumference (MUAC) < 115 mm [1]. Severe acute malnutrition is still a major public health problem in many African countries affecting the overall health and development priorities due to the effects [2, 3].

Globally, 52 million children of age less than five years were affected by acute malnutrition from which 17 million were severely acutely malnourished [1]. Most of SAM children live in South Asia and Sub-Saharan Africa [4]. Data shows that more than half of all wasted children in the world live in Southern Asia and Sub-Sahara African countries [5]. SAM is also a major cause of disability preventing children who survive from reaching their full development potentials [2, 6].

In developing countries, 2% of children suffer from severe acute malnutrition [7]. Published literatures in Africa revealed that children with SAM given Ready Used Therapeutic Feeding (RUTF) were 51% recovered than the standard care group [8]. In Ethiopia despite the improvement made in child health nutritional interventions, SAM remains in a precarious situation where under-nutrition is an underlying cause to half of its child deaths and wasting contributing to 23% of these deaths [9]. Ethiopia Demographic and Health Survey (EDHS) 2016 report showed that 38%, 10%, and 24% of under-5 years of age children in Ethiopia were stunted, wasted, and underweight respectively. In Amara region, 11.6% of under-five years of age children were wasted of which 3.5 are severely wasted [10]. Studies in Ethiopia indicates that the recovery rate among children attending the inpatient facilities was still low and the defaulter rate was high compared to the acceptable minimum standard, this has the negative impact on the child health and survival [11, 12].

Formerly in many countries, treatment of SAM had been restricted to facility-based approaches, greatly limiting its coverage and impact [13]. However, evidence from emergency programs suggested that large numbers of SAM cases could be treated in their communities

[14]. The program was started in Ethiopia in 2003 using the Community-based Therapeutic Care (CTC) model, which depended upon significant external resources and expertise and was implemented parallel to the national health system rather than integrated [15, 16]. This Community-based Management of Acute Malnutrition (CMAM) reduces the limitations of health facilities and therapeutic feeding centers (TFC) management of SAM. It addresses the community-based management of SAM children without medical complications (OTP) and Moderate Acute Malnutrition (MAM) and designed as components of CMAM [17–19].

In Ethiopia, the program has now expanded to every health center and health post of the country. OTP serve the management of SAM in children aged 6–59 months [15, 20]. The management of SAM was mainly with ready-to-use therapeutic foods (RUTF); other routine medications like antibiotics, vitamin A, and folic acid; and deworming [20, 21].

Inpatient therapeutic feeding units are faced with a lot of challenges in handling cases of severe acute malnutrition. Some of the challenges include; limited in-patient capacity, lack of enough skilled staff in the hospitals to treat the large numbers needing care, the centralized nature of hospitals promotes late presentations and high cost for treatment, increased risk of cross infections for immune-suppressed children such as children with SAM [22]. Outpatient therapeutics unit in Ethiopia context is to refer primary health care systems such as health posts, primary clinics, health centers and primary hospitals.

Besides the prevention strategies, the improved management of SAM is an integral part of the World Health Resolution on Infant and Young Child Nutrition to improve child survival. Children with SAM have profoundly disturbed physiology and metabolism when intensive re-feeding is initiated before metabolic and electrolyte imbalances corrected [23].

Despite malnutrition is being one of the major public health problems in Ethiopia, limited information exists regarding the outcome of SAM treatment provided through the outpatient decentralized approach. Even though SAM patients are being managed at OTP unites; there is scarce evidence in the efficacy of these ongoing SAM treatments in the study area. The study, therefore, is aimed to assess the treatment outcome of SAM and associated factors among under-five children in the outpatient therapeutics unit.

## Methods

### Study area and periods

The study was conducted in Gubalafto Wereda from April 2016 to May 2019 GC. Gubalafto is one of the Districts in North Wollo Zone of Amhara Region which is 521 km from Addis Ababa, North-Central Ethiopia. The total estimated population in the Woreda is 1,76,492 of whom, 90,187 male and 86,305 females. The number of children under-five years in this Wereda is 23,904. The Woreda has a total of 34 kebeles; among which 30 kebeles are rural and 4 kebeles are urban. Gubalafto Wereda administration health office report, there are 8 health centers and 34 health posts [24].

### Study design

A retrospective cohort study was conducted using document review in outpatient therapeutic feeding units of the selected health posts North Wollo Zone, Amhara region, Ethiopia, 2019.

### Population

**Source population.**   All children under-five years at outpatient therapeutic feeding units with the diagnosis of severe acute malnutrition in North Wollo Zone health posts.

**Study population.**   Under-five children at outpatient therapeutic feeding units of the selected health posts.

**Study unit.**   Medical records of the sampled children under-five years at outpatient therapeutic feeding units of the selected health posts.

## Eligibility criteria

**Inclusion criteria.**   Records of children under-five years at outpatient therapeutic feeding units.

**Exclusion criteria.**   Transferred cases and records with incomplete information were excluded.

**Sampling technique and procedure.**   A multistage sampling technique was employed to select the study subject. From the total 34 (30 rural and 4 urban) kebeles, 7 rural and 2 urban kebeles was selected by simple random sampling method.

The samples are distributed proportionally based on probability proportional to size (PPS) allocation technique. Participants in each kebele were selected by using a systematic sampling technique after calculating the sampling interval (K) for each kebeles.

The sample frame is the list of children under-five years SAM charts at OTP. It is identified after checking all 9 selected kebeles (study population) to identify charts of children from birth up to 59 months old and coding of those charts was done to prepare sampling frame for each kebele. Those children with incomplete charts are considered as non-respondent. Finally, the OTP record card of each child was selected using systematic random sampling.

**Sample size determination.**   For the first specific objective sample size for the magnitude of treatment outcome, the sample size was determined using a single population proportion formula. A study done in the OTP Wolaita zone showed a recovery rate of 64.9% and two different studies in the Amhara region showed a recovery rate of 78% and 58.4%. For this calculation, we used the proportion that was conducted in Wolaita since the two lists above done in the inpatient therapeutics unit.

$$n = \frac{(z\,\alpha/2)^2 \times p\,(1-p)}{d^2}$$

$$n = \frac{(1.96)^2 \times 0.64\,(1-0.64)}{(0.05)^2}$$

$$= 353.89$$

$$= 354$$

A total sample size of 354 was determined using single sample proportion formula by considering 95% confidence, 5% margin of error, and taking a 64.9% recovery rate from Wolaita. By adding 10% non-respondent rate the final sample size was 390

Where,

n = sample size derived from estimation formula

$Z\alpha/2$ = the value of z at a confidence level of 95% = 1.96

P = is recovery rates of children who had been managed for SAM = .64.9 (64.9%)

d = is the margin of error to be tolerated and taken as 5%

Considering 10% contingency for missing data the final sample size for determining the treatment outco

For the second objective, to assess risk factors for treatment outcome of SAM among under-five children in outpatient therapeutics unit, the sample size was determined using a

double population proportion formula by considering study was done in Tigray and Wolaita recovery rate p = 61.78 [25], 64.9 [17] respectably to calculate the required sample size. Finally, it is calculated by using Epi info version 7 statistical packages. We used Open Epi-version 2.3 [20] to calculate the sample size with the following assumptions: The proportion recovered in the exposed (children with co-morbidities) group (33.3%), the proportion recovered in the non-exposed (children without Comorbidities) group (20.4%) [21], 95% CI (confidence interval), 5% marginal error (d), and power of 80%. Accordingly, the minimum sample size calculated for each group was 374. We used a design effect of 1.5 to compensate for potential losses during multi-stage sampling and added 10% of the sample for missing and incomplete data. The final sample size obtained was 600.

$$n_1 = \frac{\left[Z_{\alpha/2}\sqrt{\left(1+\frac{1}{r}\right)P(1-P)} + Z_{\beta}\sqrt{\frac{P_1(1-P_1)+P_2(1-P_2)}{r}}\right]^2}{(P_1 - P_2)^2}$$

- **P1**: is a percent of exposed with the outcome

- **P2**: is a percent of non-exposed with the outcome

- **$Z_{\alpha/2}$**: is taking CI 95%,

- **$Z_B$**: 80% of power

- And **r** is the ratio of non-exposed to exposed 1:1

**Sampling procedure.**   The study area, Gubalafto Wereda has a total of 34 Kebeles (4 urban and 30 rural). From the total 34 kebeles, 7 rural (03, 07, 11, 14, 18, 22, 27) and 2 urban kebeles (02 and 04) were selected by simple random sampling method.

The samples are distributed proportionally based on probability proportional to size (PPS) allocation technique considering number of children under-five years SAM charts at OTP. Participants in each kebele are selected by using a systematic sampling technique after calculating the sampling interval (K) for each kebele separately.

The sample frame is the list of children under-five years SAM charts at OTP. It is identified after checking all 9 selected kebeles (study population) to identify charts of children from birth up to 59 months old and coding of those charts was done to prepare sampling frame for each kebele. Those children with incomplete charts are considered as non-respondent. Finally, the OTP record card of each child was selected using systematic random sampling.

## Study variables

**Dependent variables.**   Treatment outcome Recovered or Not Recovered
**Independent variables.**

- Socio-demographic variables: (Age, and sex)

- Type of malnutrition (Marasmus, kshiorkor and marasmic-kwashiorkor)

- Medical co-morbidities (TB, HIV, cough or pneumonia, fever, diarrhea, immunization, measles vaccine, vitamin A, routine medications, amoxicillin, folic acid, albendazole or mebendazole)

- Admission category (new, and re-admission)

**Operational definitions.**

1. Treatment outcome: grouped as recovered and not recovered from SAM management at outpatient therapeutic feeding units in this study.

2. Recovered: children with severe acute malnutrition declared as cured or recovered in the logbook of outpatient therapeutic feeding units.

3. Not recovered: defined as children discharged from outpatient therapeutic feeding units with outcome other than recovery in this study (death, default, and non-responder).

4. Severe acute malnutrition (SAM): the weight-for-height ratio of less than minus 3 standard deviations below the median WHO growth standards or weight-for-height ratio of below 70% of the median NCHS reference or presence of nutritional edema.

5. Outpatient Management: Management letter of SAM of children without medical complications or pass appetite test.

6. Defaulter: A SAM patient who becomes absent continuously from the therapeutic feeding program of outpatient care.

7. Non-responder: SAM patient admitted to inpatient that does not reach discharge criteria after 40 days in the inpatient program.

8. Died: Severe Acute Malnutrition Patient in OTP as died.

9. Type of malnutrition: grouped as marasmus (non-edematous), kwashiorkor (edematous), marasmus kwashiorkor (both edema and severe wasting), and visible severe wasting.

**Data quality control.** The data collectors and the supervisors were trained for two days on techniques of data collection and the importance of disclosing the possible purposes of the study to the study participants before the start of data collection. To assure the quality of the data, investigators closely supervised the data collection procedure daily. The review was made in the field for checking the completeness of questionnaire and correction was made in the field.

Each questionnaire and data sheet was check before the data entry. The data was entered one I data version 4.2 daily, basis and missing data were identified. Incorrectly filled or questioners that miss major content was not included in the study. The pretest was conducted in the Woldia health center (which is not a study area) using 5% of the total sample size which is not included in the actual sampling and necessary adjustments were made on the tool.

## Data processing and analysis

The data was entered and analyzed by using EPI-data Version 4.2 and exported to SPSS version 24.0 for analysis. Bivariate and Multivariate regression was also carried out to determine the association between dependent and independent variables.

## Ethical approval and consent to participate

Ethical approval was obtained from the research ethics committee of the Woldia University College of health science. An official letter of permission was obtained from Woldia University College of health science and was submitted to the respective administrative bodies of the Gubalafto Woreda; permission from these administrative bodies was also given. All records were fully anonymized before we accessed them and the ethics committee waived the

requirement for informed consent. Confidentiality was ensured throughout the research process. All incomplete charts were considered as non-response rates.

## Results

### Socio-demographic characteristics of children

The study included 600 eligible children who had been managed for SAM under the OTP from April to May (2016–2019); 50.8% of children enrolled in the study were males. Children beyond two years of age, 179 (29.8%), were underrepresented in the OTP as compared to their middle age groups, 313 (52.2%). About 18% of the children were younger. The median weight at admission, marasmic, marasmic kwashiorkor and kwashiorkor patients marasmic kwashiorkor and kwashiorkor patients were 7.7 kg (IQR: 6.2 to 10.5 kg), 7.1 kg (IQR: 5.8–9.2 kg), 8.4 kg (IQR: 7.1–9.8 kg), and 9.97 kg (IQR: 8.15–11.60 kg) respectively. Concerning vaccination history: 444 (74.0%) were fully vaccinated, 83 (13.8%) were partially vaccinated, 40 (6.7%) unknown vaccination status, and 33 (5.5%) were not vaccinated for age. The majority (90.2%) of children were identified as newly admitted children. The length of stay for severely malnourished children admitted in the OTP program ranges from 2 months up to 4 months.

Regarding treatment outcome 65% recovered, 2% were dead (Table 1).

Regarding the type of Malnutrition at Admission about 451 (75.2%) of children admitted to OTP had non-edematous (marasmic), type of severe acute malnutrition at admission (visit), (149) 24.8% of the children were kwashiorkor.

### Co-morbidity at admission

Forty present (40%) of children admitted to OTP had a fever. In some case, there were multiple comorbidities at admission including diarrhea (12.8), HIV positive 12(2%), TB 12(2%), cough 19.2% and vomiting 25.3%. In addition to this, 26.4% of the children had edema (Table 2).

### Routine medications

Admitted cases with severe acute malnutrition to OTP were managed following the federal ministry of health of Ethiopia guideline protocol for the treatment of severe acute malnutrition. Out of 600 children whose medication records were available for review, the most prescribed medications were PO antibiotics (90%) Amoxicillin followed by Vitamin A supplementation (74.0%). Of the total 44.6% of the children was dewormed with Albendazole or Mebendazole, 52.2% received folic acid (Table 3).

### Treatment outcome of SAM and associated factors

In the bivariate logistic regression analysis of malnutrition, the presence of fever, presence diarrhea, presence cough, presence vomiting, presence edema, PO antibiotics, admission category, immunization status, the weight of child were associated with treatment outcome of SAM. Those variables that have a p-value less than or equal to 0.25 were entered into a multivariable logistic regression model to adjust for possible confounders.

In a multivariate logistic regression presence of cough, the presence of diarrhea PO antibiotics, admission category, and the immunization status of a child be significantly associated with the treatment outcome of SAM. Accordingly, the odds of recovery on SAM among children presenting with cough [AOR = 0.47,95% CI: (0.28–0.80)] was lower compared to those without cough. The odds of recovery on SAM were higher among children who took PO antibiotics [AOR = 3.38, 95% CI: (1.61–7.08)] compared to those who were not taken. The admission category was also associated with the recovery of SAM. Those who have new admission

**Table 1. Socio-demographics and related characteristics of children from birth up to 59 months in Gubalafto Wereda, Ethiopia, from April 2016 to May 2019 GC (N = 600).**

| Socio-demographic and related variables | | Frequency (N = 600) | Percent |
|---|---|---|---|
| **Age** | <6 month | 108 | 18 |
| | 6–24 month | 313 | 52.2 |
| | >24 months | 179 | 29.8 |
| **Sex** | Male | 305 | 50.8 |
| | Female | 295 | 49.2 |
| **Immunization status** | Vaccinated for age | 444 | 74.0 |
| | Partially vaccinated | 83 | 13.8 |
| | Not vaccinated | 33 | 5.5 |
| | Unknown | 40 | 6.7 |
| **Season** | Winter | 142 | 23.7 |
| | Spring | 189 | 31.5 |
| | Summer | 167 | 27.8 |
| | Autumn | 102 | 17.0 |
| **Weight** | < = 6.5 | 228 | 38.0 |
| | >6.5 | 372 | 62.0 |
| **Admission category** | New | 540 | 90.2 |
| | Readmission | 59 | 9.8 |
| **Temperature** | < = 38 | 491 | 81.8 |
| | >38 | 109 | 18.2 |
| **Respiratory rate** | 30–40 | 468 | 78.1 |
| | 40–50 | 131 | 21.9 |
| **Treatment outcome** | Cured | 390 | 65.0 |
| | Dead | 12 | 2.0 |
| | Defaulter | 96 | 16.0 |
| | Medical transfer | 102 | 17.0 |

**Table 2. Co-morbidity at admission of children from birth up to 59 months in Gubalafto Wereda, Ethiopia, from April 2016 to May 2019 GC (N = 600).**

| Characteristics | Category | Frequency | Percent (%) |
|---|---|---|---|
| **Child HIV status** | Negative | 475 | 79.2 |
| | Positive | 12 | 2 |
| | Unknown | 113 | 18.8 |
| **Presence of TB** | Yes | 12 | 2 |
| | No | 482 | 80.3 |
| | Unknown | 106 | 17.7 |
| **Presence of fever** | yes | 250 | 41.7 |
| | No | 350 | 58.3 |
| **Presence of cough** | yes | 115 | 19.2 |
| | No | 485 | 80.8 |
| **Presence diarrhea** | Yes | 77 | 12.8 |
| | No | 523 | 87.2 |
| **Presence of vomiting** | Yes | 152 | 25.3 |
| | No | 448 | 74.7 |
| **Presence of edema** | **Yes** | 158 | **26.4** |
| | No | 441 | 73.6 |

**Table 3. Routine medication given at OTP for SAM children from birth up to 59 months in Gubalafto Wereda, Ethiopia, from April 2016 to May 2019 GC (N = 600).**

| Routine medication | | Frequency (N = 600) | Percent |
|---|---|---|---|
| Antibiotic/s (PO) | Yes | 540 | 90 |
| | No | 60 | 10 |
| Vit A | Yes | 444 | 74 |
| | No | 156 | 26 |
| Folic acid | Yes | 313 | 52.2 |
| | No | 287 | 47.8 |
| Albendazole or Mebendazole | Yes | 267 | 44.6 |
| | No | 332 | 55.4 |

were 3.78 times more likely to recover than those who have readmission. The odds of recovery on SAM among children presence with diarrhea [AOR = 0.46, 95% CI: (0.25–0.86)] less likely recover than those SAM children without diarrhea. Concerning immunization status of children the odds of recovery on SAM management among children who have been fully vaccinated (AOR = 6.85, 95% CI: 3.68–12.78)) was higher compared to those who have not been vaccinated (Table 4).

## Discussion

The study was mainly aimed to report on treatment outcomes of SAM in OTP and associated factors with it among children treated with SAM. Accordingly, the overall prevalence of cured, dead, defaulter, and medical transfer were 65.0, 2.0, 16.0, and 17.0 respectively. The result revealed 600 (65%) SAM children admitted to OTP were recovered. This indicates the recovery rate was lower than the sphere standard acceptable range [26]. The finding was also lower compared to 76.8 and 80% recovery rates from the study done in OTP Tigiray and Zambia. However, it was comparable to 64.9 and 62.4% rates from a study done in Wolaita Zone [17, 21]. The disparities in reports might be due to difference in settings where SAM management was carried out. This study finding is higher than the previous study done in Nigeria 58%. The possible reason for this discrepancy might be the increased Ethiopian Government efforts to improve maternal and child nutrition through a community-based Health Extension Program and variation in the study setting.

The overall defaulter rate in this study in line with study in Tigray 17.5% [25]. This finding is higher than the study finding from Wolaita [17, 21]. This discrepancy might be due to the increased emphasis to community based therapeutic feeding program. Marasmus was found the predominant form of malnutrition in this study (75.2%), which is in line with the study done at Tigray region (98.4%) and Wolaita (63.4) [17, 21]. This may be explained by the fact that marasmus is more common in the age group below two years which is the case in this study in which 70.2% of the study population lies in the age category less-than two years. Regarding the death rate, this study reported a lower proportion of death (2%) compared to previous findings in Wolaita and Tigray region [17, 21, 25]. It was also lower than the recommended minimum sphere standard which should be <10%. In this study, the proportion of death was lower. This is mainly could be children reach OTP early before developing complications. The other reason may be appropriate management of children such as prescription of routine medication.

Children provided with amoxicillin were 3.38 times more likely to recover when compared to their counterparts. This result was consistent with the finding from North Ethiopia [21]. This can be explained by the supportive effect of amoxicillin in the treatment infections and

**Table 4. Treatment outcome of SAM and associated factors among under-five children in the outpatient therapeutics unit in Gubalafto Wereda, Ethiopia, from April 2016 to May 2019 GC G.C.**

| Variables | Treatment outcome (N = 600) | | Odds Ratios | | |
| --- | --- | --- | --- | --- | --- |
| | Recovered frequency (%) | Not recovered Frequency (%) | COR(95% CI) | AOR(95% CI) | P-value |
| Type of malnutrition | | | | | |
| Marasmus | 318(53.0) | 133(22.2) | 2.56(1.75–3.74) | 1.71(.72–4.08) | .22 |
| Kwashiorkor | 72(12.0) | 77(12.8) | | | |
| Presence of TB | | | | | |
| Yes | 5(0.8%) | 7(1.2%) | .34(.10–1.14) | .70(.16–2.99) | .63 |
| No | 313(52.2%) | 169(28.2%) | .88(.56–1.37) | 1.10(.64–1.89) | .73 |
| Unknown | 72(12.0%) | 34(5.7%) | 1 | | 1 |
| Presence of fever | | | | | |
| Yes | 150(25.0) | 100(16.7) | 0.68(0.49–0.96) | 1.06(.69–1.63) | .78 |
| No | 240(40.0) | 110(18.3) | 1 | 1 | |
| Presence of cough | | | | | |
| Yes | 49(8.2) | 66(11.0) | 0.31(0.21–0.48) | .47(28-.80) | .005 |
| No | 341(56.8) | 144(24.0) | 1 | 1 | |
| Presence of diarrhea | | | | | |
| Yes | 31(5.2) | 46(7.7) | 0.31(0.19–0.50) | .46(.25-.86) | .015 |
| No | 359(59.8) | 164(27.3) | 1 | 1 | |
| Presence of vomiting | | | | | |
| Yes | 79(13.2) | 73(12.2) | 0.47 (0.33–0.70) | .71(.43–1.17) | .18 |
| No | 311(51.8) | 137(22.8) | 1 | 1 | |
| Presence of edema | | | | | |
| Yes | 79(13.2) | 79(13.2) | 0.42(0.29–0.61) | 1.24(.53–2.93) | .62 |
| No | 311(51.9) | 130(21.7) | 1 | 1 | |
| Child take vit A | | | | | |
| Yes | 298(49.7) | 146(24.3) | 1.42(.98–2.07) | 1.20(.76–1.90) | .433 |
| No | 92(15.3) | 64(10.7) | 1 | 1 | |
| PO antibiotics | | | | | |
| Yes | 361(60.2) | 179(29.8) | 2.16(1.26–3.69) | 3.38(1.61–7.08) | .001 |
| No | 29(4.8) | 31(5.2) | 1 | 1 | |
| Admission category | | | | | |
| New | 376(62.8) | 164(27.4) | 7.33(3.91–13.71) | 3.78(1.77–8.07) | .001 |
| Re-admission | 14(2.3) | 45(7.5) | | | |
| Immunization status | | | | | |
| Vaccination for age | 343(57.2) | 101(16.8) | 10.38 (5.83–18.47) | 6.85(3.68–12.76) | < .001 |
| Partially vaccinated | 29(4.8) | 54(9.0) | 1.64(.82–3.30) | 1.26(.60–2.67) | .54 |
| Not vaccinated /unknown | 18(3.0) | 55(9.2) | 1 | 1 | |
| Weight of the child | | | | | |
| ≤ 6.5kg | 81(13.5%) | 147(24.5%) | .55(.39-.78) | 1.08(.70–1.65) | .734 |
| >6.5 kg | 129(21.5%) | 243(40.5%) | 1 | 1 | |

other complications associated with SAM [20]. All children might not show clinical signs and symptoms of systematic infections as a result of their low body immunity.

Regarding factor associated with SAM treatment outcome (recovered/not recovered) this study revealed presence of cough, the presence of diarrhea, PO antibiotics, admission category, and the immunization status of a child as predictor of SAM treatment outcome. Presence of complications like cough and diarrhea were 53% less likely compared to their counterparts.

This might be due to the fact that coughs and diarrhea are usual associated with infections which result in further nutritional consumption which shares nutrients for nutritional recovery. Children who were fully and partially vaccinated had approximately 7 times better recovery rate when compared to those who have been not vaccinated which is almost consistent with a study in Bahirdar Felege Hiwot hospital of 4.4 times [12]. The most probable reason for this is that immunization against microorganism that causes disease can prepare the body immune system, thus helping to fight or prevent infection and an easier way to become immune to a particular disease. The odds of recovery on SAM were nearly 2 times higher among children who took PO antibiotics compared to counterparts. The possible justification for this might be PO antibiotics treat asymptomatic infections which impede the progress of SAM treatment outcome. Regarding the admission category, the current study revealed that approximately 4 times higher odds of recovery were observed among newly admitted children as compared to those who readmitted. This could be explained as children whose readmitted come with more complications, which ultimately decreases the recovery rate.

## Conclusions

Recovery rates in the study area are below the cut of points of the minimum standard sets in humanitarian and disaster prevention (or the sphere standards), it is low when compared to similar studies conducted in different parts of Ethiopia but the death rate was lower than the international standard. Presences of cough comorbidities were statistically significant factors that hinder the recovery rate of malnourished children. On the other hand, vaccination and administration PO antibiotics were positive indicators for recovery. Attachment of follow-up chart to the individual folder and monitoring of the child progress with the chart also has a greater contribution in improving the recovery of children with severe acute malnutrition in the TFU. Thus, the health care providers should emphasize those SAM cases with comorbidity like cough and readmission case which require strict follow up according to the protocol and increased use of SAM management follow up chart for all SAM patients. It is also recommended to give community-based health education and counseling for mothers to enhance child immunization.

## Acknowledgments

First of all, we would like to extend our thanks to Woldia University Faculty of Health Sciences, Department of Nursing for providing us the golden opportunity to carry out this study. Also, we would thank the North Wollo Zone and Gubalafto Wereda Administrative Health Office for their cooperation for giving information related to the general information about the stud area and study population.

## Author Contributions

**Conceptualization:** Biruk Beletew Abate, Befkad Deresse Tilahun, Ayelign Mengesha Kassie, Mesfin Wudu Kassaw.

**Data curation:** Biruk Beletew Abate.

**Formal analysis:** Biruk Beletew Abate, Befkad Deresse Tilahun, Ayelign Mengesha Kassie.

**Funding acquisition:** Biruk Beletew Abate, Befkad Deresse Tilahun, Ayelign Mengesha Kassie.

**Investigation:** Biruk Beletew Abate, Befkad Deresse Tilahun, Mesfin Wudu Kassaw.

**Methodology:** Biruk Beletew Abate, Befkad Deresse Tilahun, Mesfin Wudu Kassaw.

**Project administration:** Biruk Beletew Abate, Ayelign Mengesha Kassie, Mesfin Wudu Kassaw.

**Resources:** Biruk Beletew Abate, Ayelign Mengesha Kassie.

**Software:** Biruk Beletew Abate, Ayelign Mengesha Kassie, Mesfin Wudu Kassaw.

**Supervision:** Biruk Beletew Abate, Mesfin Wudu Kassaw.

**Validation:** Biruk Beletew Abate.

**Visualization:** Biruk Beletew Abate.

**Writing – original draft:** Biruk Beletew Abate, Befkad Deresse Tilahun, Mesfin Wudu Kassaw.

**Writing – review & editing:** Biruk Beletew Abate, Ayelign Mengesha Kassie.

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
