## [Decision Letter · Decision Letter 0]

7 May 2020

PONE-D-19-29446

Treatment Outcome of Severe Acute Malnutrition and associated factors among under-five children in outpatient therapeutics unit in Gubalafto Wereda, North Wollo Zone, Amara Regional State, North Ethiopia, 2019 G.C

PLOS ONE

Dear Mr Beletew,

Thank you for submitting your manuscript to PLOS ONE. After careful consideration, we feel that it has merit but does not fully meet PLOS ONE’s publication criteria as it currently stands. Therefore, we invite you to submit a revised version of the manuscript that addresses the points raised during the review process.

You can see that a large revision is necessary and the suggestions must be responded entirely,

It is necessary a complete English revision after your changes have be done, for the entire manuscript. The referees have pointed some of them, but there are even more. Please use a professional native English translator. 

We would appreciate receiving your revised manuscript by Jun 21 2020 11:59PM. To enhance the reproducibility of your results, we recommend that if applicable you deposit your laboratory protocols in protocols.io, where a protocol can be assigned its own identifier (DOI) such that it can be cited independently in the future. For instructions see: http://journals.plos.org/plosone/s/submission-guidelines#loc-laboratory-protocols

We look forward to receiving your revised manuscript.

Kind regards,

Ricardo Q. Gurgel, PhD

Academic Editor

PLOS ONE

Journal Requirements:

https://jhpn.biomedcentral.com/articles/10.1186/s41043-017-0083-3

https://journals.plos.org/plosone/article?id=10.1371%2Fjournal.pone.0065840

In your revision ensure you cite all your sources (including your own works), and quote or rephrase any duplicated text outside the methods section. Further consideration is dependent on these concerns being addressed.

4. In ethics statement in the manuscript and in the online submission form, please provide additional information about the patient records used in your retrospective study.

Specifically, please ensure that you have discussed whether all data were fully anonymized before you accessed them and/or whether the IRB or ethics committee waived the requirement for informed consent.

If patients provided informed written consent to have data from their medical records used in research, please include this information.

5. Thank you for including your ethics statement:

"Ethical approval was obtained from the research ethics review board of the WU faculty of health science."

7. Thank you for stating the following financial disclosure: 'N/A'

8. Please amend the manuscript submission data (via Edit Submission) to include author Befkad Adresse.

9. Please amend your authorship list in your manuscript file to include author Befkad Deresse.

10. Your ethics statement must appear in the Methods section of your manuscript. If your ethics statement is written in any section besides the Methods, please move it to the Methods section and delete it from any other section. Please also ensure that your ethics statement is included in your manuscript, as the ethics section of your online submission will not be published alongside your manuscript.

Reviewers' comments:

Reviewer's Responses to Questions

**Comments to the Author**

1. Is the manuscript technically sound, and do the data support the conclusions?

Reviewer #1: Yes

Reviewer #2: No

2. Has the statistical analysis been performed appropriately and rigorously? 

Reviewer #1: Yes

Reviewer #2: No

3. Have the authors made all data underlying the findings in their manuscript fully available?

Reviewer #1: Yes

Reviewer #2: Yes

4. Is the manuscript presented in an intelligible fashion and written in standard English?

Reviewer #1: Yes

Reviewer #2: No

5. Review Comments to the Author

Reviewer #1: I have attached a document with recommended changes which are almost all relating to editorial changes. Please see attached and once these editorial changes are made, the content is appropriate.

I would bring the lower death rates observed into both the abstract and conclusions. This is valuable and commendable work and I hope the grammatical changes I suggested are not too painful to implement.

Reviewer #2: Comments on

PONE-D-19-29446: Treatment Outcome of Severe Acute Malnutrition and associated factors among under-five children in outpatient therapeutics unit in Gubalafto Wereda, North Wollo Zone, Amara Regional State, North Ethiopia, 2019 G.C

Major Comments:

1. The title shall be re-written as “Treatment Outcome of Severe Acute Malnutrition and associated factors among under-five children in outpatient therapeutics unit in Gubalafto Wereda, North Wollo Zone, Ethiopia”

2. The Background section requires major revision. The first paragraph of the background section shall start by defining “Severe acute malnutrition”. The next paragraph shall describe the burden of severe acute malnutrition in the globe, in Africa, in Ethiopia and the study area. The ways of preventing and treating severe acute malnutrition and respective outcomes shall be presented with corresponding references.

3. There is also a need to define “outpatient therapeutics unit” in the Ethiopian context. Does it refer to health care systems (health posts, primary clinic, primary hospital etc)

4. The justification for the study “Besides, the high percentage of malnutrition is alarming which needs further study to describe the treatment outcome of SAM in OTP to assess the factors contributing to the treatment outcome.” is not adequate. It rather shall state whether there is an ongoing SAM treatment and hence evidence of gap in the efficacy of the ongoing SAM treatment in the study area. The justification as it stands now is vague to warrant the objective. Hence there is a need for major revision.

5. The Objective “The study, therefore, is aimed at describing the treatment outcome among children of age less than five years and identifies factors contributing to the treatment outcome.” is not congruent with the title “Treatment Outcome of Severe Acute Malnutrition and associated factors ………….” which is quite important to revise

6. Several typographic, grammatical and logical errors are rampant in the background section and need to be corrected

7. Methods: There is repetition on the sampling technique and procedure. “The study area, Gubalafto Wereda has a total of 34 Kebeles (4 are urban and 30 are rural kebeles). From the total 34 kebeles, 7 rural and 2 urban kebeles was selected by simple random sampling method” is repeated. The sample size is different (390, 374, 600). And the reason how you came up with the total samples of 600 is not clear yet. The names of the selected Kebeles shall be presented here. To which objective does the statement “For the second objective, the sample size was determined using a double population proportion formula by considering study done in Tigray and Wolaita recovery rate p=61.78,64.9 respectably to calculate the required sample size. Finally, it is calculated by using Epi info version 7 statistical packages.” correspond? If the authors have two specific objectives they shall present them clearly in the background section and also state the sample size to each objective clearly. In addition, each specific objective shall have a clear background and/or justification in the background section. The statement “Participants in each kebele are selected by using a systematic sampling technique after calculating the sampling interval (K=2) for each kebeles” sounds vague as the number of residents, population etc varies across the kebeles and hence the corresponding sampling interval. Hence this section shall be clearly presented as it is central to the study.

8. Several typographic errors in the methods section require major revision.

9. Results section: The results shall show the kebeles/health posts and corresponding cases. This section shall present the results clearly. Statistical figures ( e.g. differences in the mean values of variables are not presented under the section “Socio-demographic characteristics of children” which is important for meaningful comparisons and extrapolations. Tables 1, 2, 3, and 4 do not show the time frame or the period (eg. September 2000-August 2004) of the data to inform readers on matters such as on the magnitude of cases occurred in the area.

10. The number(%) of severe cases and mild cases should be presented clearly indicating the totoal cases, and the cases in each kebele.

11. Results: the section “Bivariate and Multivariate analysis on treatment outcome of SAM and associated factors” shall be re-written as “Treatment outcome of SAM and associated factors”

12. Under the section “Treatment outcome of SAM and associated factors” it remains important to present the length of treatment and related outcome. As it has several advantages for policy and follow up.

13. “Table 4: Bivariate and Multivariate analysis on treatment outcome of SAM and associated factors” shall be re-written as “Table 4: Treatment outcome of severe acute malnutrition and associated factors in ------ Woreda, Northern Wollo, Ethiopia ---month, ----year to -----month, ----year”

14. Discussion: This section shall present the main findings obtained by analyzing the data in line with the objective and discuss contextually. “The study was mainly aimed to indicates treatment outcomes of OTP and associated factors with it among children treated from SAM” This is vague and requires clarification. Did you address “Accordingly, the overall prevalence of cured, dead, defaulter, and medical transfer were 65.0, 2.0, 16.0, and 17.0 respectively” in your study adequately

6. PLOS authors have the option to publish the peer review history of their article (what does this mean?). If published, this will include your full peer review and any attached files.

Reviewer #1: Yes: Marie G Chivers

Reviewer #2: No

---

## [Author Response · Author response to Decision Letter 0]

15 May 2020

Date: May 15/2020

To: "PLOS ONE" plosone@plos.org

From: "Biruk Beletew" birukkelemb@gmail.com

Subject: Submitting Revision Version [PONE-D-19-29446]

PONE-D-19-29446

Treatment Outcome of Severe Acute Malnutrition and associated factors among under-five children in outpatient therapeutics unit in Gubalafto Wereda, North Wollo Zone, Amara Regional State, North Ethiopia, 2019 G.C

PLOS ONE

Dear Editor we have no word to explain our deepest thanks for your constructive comments and helping us throughout the process of preparing the manuscript for publication. Since we have agreed with all points you raised we believe we have carefully amended the paper as per your point of view and the journal guideline. We have downloaded the journal guideline and prepare the manuscript accordingly. Thank you again since you have contributed much for our better paper.

We have presented below the point to point response to each pointes raised by the editor and the reviewers.

 Editor comment: 1. Please ensure that your manuscript meets PLOS ONE's style requirements, including those for file naming. The PLOS ONE style templates can be found at

Authors’ response: we have used the attached documents as a guide to prepare the manuscript and really helped us. 

 Editor comment: 2. We suggest you thoroughly copyedit your manuscript for language usage, spelling, and grammar. If you do not know anyone who can help you do this, you may wish to consider employing a professional scientific editing service. 

 Editor comment: 3. We noticed you have some minor occurrence of overlapping text with the following previous publication(s), which needs to be addressed:

https://jhpn.biomedcentral.com/articles/10.1186/s41043-017-0083-3

https://journals.plos.org/plosone/article?id=10.1371%2Fjournal.pone.0065840

In your revision ensure you cite all your sources (including your own works), and quote or rephrase any duplicated text outside the methods section. Further consideration is dependent on these concerns being addressed.

 Authors’ response: we have paraphrased the manuscript and removed the textual overlap. We have also cited those published works in method part.

Editor comment: 4. In ethics statement in the manuscript and in the online submission form, please provide additional information about the patient records used in your retrospective study.

Specifically, please ensure that you have discussed whether all data were fully anonymized before you accessed them and/or whether the IRB or ethics committee waived the requirement for informed consent. If patients provided informed written consent to have data from their medical records used in research, please include this information.

Authors’ response: we have amended the ethics statement as per your comment as “ All records were fully anonymized before we accessed them and the ethics committee waived the requirement for informed consent. Confidentiality was ensured throughout the research process. All incomplete charts were considered as non-response rate.” (Page 10 Line 244-246)

 Editor comment: 5. Thank you for including your ethics statement:

"Ethical approval was obtained from the research ethics review board of the WU faculty of health science."

 Authors’ response: we have amended as per your comment as “Ethical approval was obtained from the research ethics committee of the Woldia University College of health science. An official letter of permission was obtained from Woldia University College of health science and was submitted to the respective administrative bodies of the Gubalafto woreda; permission from these administrative bodies was also given. All records were fully anonymized before we accessed them and the ethics committee waived the requirement for informed consent. Confidentiality was ensured throughout the research process. All incomplete charts were considered as non-response rate.” (Page 10 Line 241-247)

 Editor comment: 6. In your Data Availability statement, you have not specified where the minimal data set underlying the results described in your manuscript can be found. PLOS defines a study's minimal data set as the underlying data used to reach the conclusions drawn in the manuscript and any additional data required to replicate the reported study findings in their entirety. All PLOS journals require that the minimal data set be made fully available. For more information about our data policy, please see http://journals.plos.org/plosone/s/data-availability.

 Authors’ response: we have indicated the availability of data and materials as “All relevant data are within the paper”:page18; Line 375

Editor comment: 7. Thank you for stating the following financial disclosure: 'N/A'

a. Please clarify the sources of funding (financial or material support) for your study. List the grants or organizations that supported your study, including funding received from your institution.

d. If you did not receive any funding for this study, please state: “The authors received no specific funding for this work.”

 Authors’ response: we have amended the manuscript as per the comment: Funding: The study was funded by Woldia University. However, the funder had no role in study design, data collection and analysis, decision to publish, or preparation of the manuscript:Page 18 line 379-380

Editor comment: 8. Please amend the manuscript submission data (via Edit Submission) to include author Befkad Adresse.

 Authors’ response: sorry it was by mistake, the correct one is Befkad Deresse, and it have been amended in both editorial system and the manuscript: Page 1 Line 5

Editor comment: 9. Please amend your authorship list in your manuscript file to include author Befkad Deresse.

Authors’ response: sorry it was by mistake, the correct one is Befkad Deresse, and it have been amended in both editorial system and the manuscript: Page 1 Line 5

 Editor comment: 10. Your ethics statement must appear in the Methods section of your manuscript. If your ethics statement is written in any section besides the Methods, please move it to the Methods section and delete it from any other section. Please also ensure that your ethics statement is included in your manuscript, as the ethics section of your online submission will not be published alongside your manuscript.

Authors’ response: adjusted the ethics statement appear in the Methods section not elsewhere (Page 10 Line 241-247)

 To Reviewer #1

Dear Reviewer we would like to forward our deep-seated gratitude for your interesting and valuable comments and helping us throughout the process. We all really appreciate your potential and optimism while you give such constructive, thoroughly and in-depth comments. Since we have agreed with all of your points raised we have amended the manuscript as per your comments. We would like to thank you again since you are contributing for our better paper by giving such comments which are important to improve the quality of this paper. Below we have written the point to point response to issues you raised.

Reviewer comment: I have attached a document with recommended changes which are almost all relating to editorial changes. Please see attached and once these editorial changes are made, the content is appropriate.

I would bring the lower death rates observed into both the abstract and conclusions. This is valuable and commendable work and I hope the grammatical changes I suggested are not too painful to implement.

Authors’ response: thank you very much, we have listed your comments taken from the attached file and indicated as we have amended and highlighted in the manuscript.

Abstract

• Reviewer comment: Replace evidence on with evidence of

• Authors’ response: amended as per the comment: Page 1 Line 18

• Reviewer comment: define the word and then follow with (SAM) and for OTP

• Authors’ response: amended as per the comment: Page 2 Line 26

• Reviewer comment: the recovery rate was revealed as 65 % with the recovery rate was found to be 65%

• Authors’ response: amended as per the comment: Page 2 Line 34

• Reviewer comment: Replace Children who took immunization were had 6.85 times with Immunized children had 6.85..

• Authors’ response: amended as per the comment: Page 2 Line 35

• Reviewer comment: Replace recover compared to their counterparts who were not provided with recover than their counterparts

• Authors’ response: amended as per the comment: Page 2 Line 39

• Reviewer comment: Replace with treatment outcome of Sever Acute Malnutrition With-with the treatment outcome of severe acute malnutrition.

• Authors’ response: amended as per the comment: Page 2 Line 46

• Reviewer comment: make a stronger link in the concluding statement between the outpatient services existing and the impact - e.g. The impact on the recovery rates of children treated using the OTP service indicate the potential benefits of increasing the capacity of such services across a target region on child mortality/recovery. The importance of timely intervention is another benefit of a more local service.

• Authors’ response: Thank you very much, we have amended as per the comment. Page 2 Line 47-51.

Background

• Reviewer comment: Across the document define abbreviations on first time of use 

• Authors’ response: we have defined abbreviations on first time of use throughout the document 

• Reviewer comment: Replace the program now with the program has now 

• Authors’ response: amended as per the comment: Page 4 Line 87

• Reviewer comment: I am not sure I understand the second part of this sentence "and high opportunity cost for careers"…. Page 4 Line 94-95

• Authors’ response: amended as per the comment

• Reviewer comment: Replace metabolism, such that if intensive re-feeding metabolism with when intensive refeeding...

• Authors’ response: amended as per the comment: Page 4 Line 100-101

• Reviewer comment: Replace Despite malnutrition is one with despite malnutrition being

• Authors’ response: amended as per the comment: Page 4 Line 102

Methods 

• Reviewer comment: Methods title needs to be aligned with the methods section

• Authors’ response: amended as per the comment: Page 5 Line 110

• Reviewer comment: one of the Wereda x.. provinces/districts?

• Authors’ response: amended as per the comment: Page 5 Line 113

• Reviewer comment: delete "found at"

• Authors’ response: deleted as per the comment

• Reviewer comment: delete far

• Authors’ response: deleted as per the comment

• Reviewer comment: whom 80,187 male and 86,305 females

• Authors’ response: amended as per the comment: 

• Reviewer comment: children under five years

• Authors’ response: amended as per the comment : Page 5 Line 115

• Reviewer comment: All children under five years - across document 

• Authors’ response: amended as per the comment

• Reviewer comment: will be excluded

• Authors’ response: amended as was excluded: Page 5 Line 135

• Reviewer comment: in this paragraph replace a proportion that was done with "a proportion that was conducted in xx" and keep in the past tense. We used..

• Authors’ response: amended as per the comment

• Reviewer comment: small c - co-morbidities

• Authors’ response: amended as per the comment

• Reviewer comment: Create table or put in header titles with bullets under them; or titles followed by :x,y,z 

• Authors’ response: amended as per the comment: Page 6 Line 152-153

• Reviewer comment: cured or recovered

• Authors’ response: amended as per the comment: Page 9 Line 212

• Reviewer comment: space after the number 179 (29.8%) across all document

• Authors’ response: amended as per the comment: Page 10 Line 255

• Reviewer comment: in the table Make first letter of all words on list capital –

• Authors’ response: amended as per the comment: Page 11 Table 1

• Reviewer comment: In some cases there were multiple co-morbidities at admission to include –

• Authors’ response: amended as per the comment

• Reviewer comment: In bivariate logistic regression analysis of malnutrition, the presence of....

• Authors’ response: amended as per the comment: Page 13 Line 291

• Reviewer comment: In a multivariate..

• Authors’ response: amended as per the comment: Page 13 Line 291

• Reviewer comment: delete "than as"

• Authors’ response: deleted as per the comment

• Reviewer comment: delete as

• Authors’ response: deleted as per the comment

• Reviewer comment: across document put in the 0 before the decimals e.g. 0.25-0.86

• Authors’ response: amended as per the comment: Page 14 Line 304

• Reviewer comment: delete as

• Authors’ response: amended as per the comment

• Reviewer comment: not been

• Authors’ response: amended as per the comment: Page 14 Line 307

• Reviewer comment: would keep all one color, or do full rows in a single color, or highlight important parts in bold text

• Authors’ response: amended as per the comment across all tables

Discussion

• Reviewer comment: to report on the

• Authors’ response: amended as per the comment: Page 16 Line 335

• Reviewer comment: children treated with SAM

• Authors’ response: amended as per the comment: Page 16 Line 376

• Reviewer comment: and between these two values

• Authors’ response: amended as per the comment: Page 16 Line 322

• Reviewer comment: higher than study findings from

• Authors’ response: amended as per the comment: Page 16 Line 329

• Reviewer comment: death rate, this study reported a lower proportion

• Authors’ response: amended as per the comment: Page 16 Line 335

• Reviewer comment: compared to

• Authors’ response: amended as per the comment: Page 16 Line 335

• Reviewer comment: This needs to be brought out as a key findings and reported in the abstract: indicates impact of the local access point for the most severe cases or reduction of fatalities

• Authors’ response: modified as per the comment: Page 2 Line 46-51

• Reviewer comment: when not as

• Authors’ response: amended as per the comment: Page 17 Line 351

• Reviewer comment: when not as

• Authors’ response: amended as per the comment: Page 17 Line 365

• Reviewer comment: Space and then - The most probable reason for this is that immunization..

• Authors’ response: amended as per the comment: Page 17 Line 353

• Reviewer comment: for not to this

• Authors’ response: amended as per the comment: Page 17 Line 353

• Reviewer comment: decreases

• Authors’ response: amended as per the comment: Page 17 Line 362

• Reviewer comment: when

• Authors’ response: amended as per the comment: Page 17 Line 365

• Reviewer comment: were

• Authors’ response: amended as per the comment: Page 17 Line 367

• Reviewer comment: hinder the..

• Authors’ response: amended as per the comment: Page 17 Line 368

• Reviewer comment: and administration of PO

• Authors’ response: amended as per the comment: Page 17 Line 369

• Reviewer comment: require instead of which need

• Authors’ response: amended as per the comment: Page 17 Line 373

• Reviewer comment: increased

• Authors’ response: amended as per the comment: Page 17 Line 374

• Reviewer comment: follow up charts for all

• Authors’ response: amended as per the comment: Page 17 Line 374

To Reviewer #2

Dear we would like to forward our deep-seated gratitude for your interesting and valuable comments and helping us throughout the process. We all really appreciate your potential and optimism while you give such constructive, thoroughly and in-depth comments. Since we have agreed with all of your points raised we have amended the manuscript as per your comments. We would like to thank you again since you are contributing for our better paper by giving such comments which are important to improve the quality of this paper. Below we have written the point to point response to issues you raised.

Reviewer comment: 1. The title shall be re-written as “Treatment Outcome of Severe Acute Malnutrition and associated factors among under-five children in outpatient therapeutics unit in Gubalafto Wereda, North Wollo Zone, Ethiopia”

Authors’ response: amended as per your comment (Page 1, Line 1-4) 

Reviewer comment: 2. The Background section requires major revision. 

The first paragraph of the background section shall start by defining “Severe acute malnutrition”. The next paragraph shall describe the burden of severe acute malnutrition in the globe, in Africa, in Ethiopia and the study area. The ways of preventing and treating severe acute malnutrition and respective outcomes shall be presented with corresponding references.

Authors’ response: Thanks for this interesting comments. We entirely updated the background section considering your comments. First paragraph: we defined SAM, Second paragraph: burden of severe acute malnutrition in the world, Third paragraph: burden of severe acute malnutrition in Developing countries, Africa, and in Ethiopia context, Next paragraphs: we tried to indicate the previous hospital based management of SAM and its challenges; then we described the purpose of OTP over the hospital based management of SAM: Finally, we explained as there is scarcity of evidence on the treatment outcome of SAM managed it OTP despite it is being implemented in the study area context. 

Reviewer comment: 3. There is also a need to define “outpatient therapeutics unit” in the Ethiopian context. Does it refer to health care systems (health posts, primary clinic, primary hospital etc)

Authors’ response: outpatient therapeutics unit in Ethiopia context is to refer primary health care systems such as health posts, primary clinics, health centers and primary hospital.(Background page 4, line 96-97)

Reviewer comment: 4. The justification for the study “Besides, the high percentage of malnutrition is alarming which needs further study to describe the treatment outcome of SAM in OTP to assess the factors contributing to the treatment outcome.” is not adequate. It rather shall state whether there is an ongoing SAM treatment and hence evidence of gap in the efficacy of the ongoing SAM treatment in the study area. The justification as it stands now is vague to warrant the objective. Hence there is a need for major revision.

Authors’ response: yes, you raised important point we missed. SAM patients are being treated at OTP. But, there is limited data on the treatment outcome (effectiveness) of that treatment. We have amended considering your comment as: Even though SAM patients are being managed at OTP unites; there is scarce evidence in the efficacy of these ongoing SAM treatments in the study area (Background: page 4 line 104-107)

Reviewer comment: 5. The Objective “The study, therefore, is aimed at describing the treatment outcome among children of age less than five years and identifies factors contributing to the treatment outcome.” is not congruent with the title “Treatment Outcome of Severe Acute Malnutrition and associated factors ………….” which is quite important to revise

Authors’ response: very nice comment, we have amended it. “To assess treatment outcome of SAM and associated factors among under-five children in outpatient therapeutics unit” (Background: page 4 line 106-107)

Reviewer comment: 6. Several typographic, grammatical and logical errors are rampant in the background section and need to be corrected

Authors’ response: Regarding English language (typographic, grammatical and logical errors) we have consulted a native English speaking collogues and they have edited the paper. We (all authors) have also edited it through repetitive checking and online grammar editor. 

Reviewer comment: 7. Methods: There is repetition on the sampling technique and procedure. “The study area, Gubalafto Wereda has a total of 34 Kebeles (4 are urban and 30 are rural kebeles). From the total 34 kebeles, 7 rural and 2 urban kebeles was selected by simple random sampling method” is repeated. The sample size is different (390, 374, 600). And the reason how you came up with the total samples of 600 is not clear yet. The names of the selected Kebeles shall be presented here. 

Authors’ response: the redundant statement has been deleted. Regarding the sample size (600): First we calculated considering both objectives separately; the first objective is on the magnitude of treatment outcome (Recovered/not recovered) by single proportion formula, it becomes 354, the second objective factors which affect the treatment outcome by double proportion formula by taking children with co-morbidities as a factor which gives maximum sample size, then than other factors the sample size becomes 374. Then we selected the one which gives maximum sample size which was the second objective. Finally, we used a design effect of 1.5 to compensate for potential losses during multi-stage sampling and added 10% of the sample for missing and incomplete data. And the final sample size becomes 600 (Method Page 6-7)

Reviewer comment: To which objective does the statement “For the second objective, the sample size was determined using a double population proportion formula by considering study done in Tigray and Wolaita recovery rate p=61.78,64.9 respectably to calculate the required sample size. Finally, it is calculated by using Epi info version 7 statistical packages.” correspond? If the authors have two specific objectives they shall present them clearly in the background section and also state the sample size to each objective clearly. In addition, each specific objective shall have a clear background and/or justification in the background section. Authors’ response: The second objective is: to assess risk factors for treatment outcome of SAM among under-five children in outpatient therapeutics unit. The sample size for the second objective were calculated by taking P1 (proportion of recovery among exposed) and P2 (proportion of recovery among un-exposed) and using double population proportion formula for each factors. Using the second objective the sample size was 600. The first objectives were to assess the treatment outcome (recovered/ not recovered) of SAM. We used single proportion population formula. Using first objective the maximum sample size was 390. Finally from the two objective the one (the second objective) which gives maximum sample size were taken, then the sample size 600 were taken.

Reviewer comment: The statement “Participants in each kebele are selected by using a systematic sampling technique after calculating the sampling interval (K=2) for each kebeles” sounds vague as the number of residents, population etc varies across the kebeles and hence the corresponding sampling interval. Hence this section shall be clearly presented as it is central to the study.

Authors’ response: ye you are correct the sample frame, children under-five years SAM charts at OTP vary across kebeles. But what we have done were, we distributed the sample to each selected kebeles proportionally based on probability proportional to size (PPS) allocation technique considering the size of the sampling frame; That means, kebeles which have large number of children under-five years SAM charts at OTP will take larger number of the samples and vis versa. After sample distribution we used systematic sampling technique by calculating K(sampling interval) in each kebele separately. (Method page 8)

Reviewer comment: 8. Several typographic errors in the methods section require major revision.

Authors’ response: Regarding English language (typographic, grammatical and logical errors) we have consulted a native English speaking collogues and they have edited the paper. We (all authors) have also edited it through repetitive checking and online grammar editor. 

Results section

Reviewer comment: 9. Results section: The results shall show the kebeles/health posts and corresponding cases. 

Authors’ response: the kebeles are indicated in number like in rural (01-30) and in urban kebeles (01-04). By simple random sampling technique 7(03, 07, 11, 14, 18, 22, 27) rural and 2 urban kebeles (02 and 04) were selected. (Page 8 line 198)

Reviewer comment: This section shall present the results clearly. Statistical figures ( e.g. differences in the mean values of variables are not presented under the section “Socio-demographic characteristics of children” which is important for meaningful comparisons and extrapolations. 

Authors’ response: we have amended the socio-demographic characteristics of children section considering your comments. Eg. We have incorporated the median weight at admission, marasmic , marasmic kwashiorkor and kwashiorkor patients were 7.7 kg (IQR: 6.2 to10.5 kg), 7.1 kg (IQR 5.8 –9.2 kg), 8.4 kg (IQR 7.1–9.8 kg), and 9.97 kg (IQR 8.15–11.60 kg) respectively. (Page 10 line 257-259)

Reviewer comment: Tables 1, 2, 3, and 4 do not show the time frame or the period (eg. September 2000-August 2004) of the data to inform readers on matters such as on the magnitude of cases occurred in the area.

Authors’ response: we have included the study period at the title of each tableincluding the period from April 2016 to May 2019 GC.

Reviewer comment: 10. The number (%) of severe cases and mild cases should be presented clearly indicating the total cases, and the cases in each kebele.

Authors’ response: since our study is conducted in OTP, all cases are diagnosed with Sever Acute Malnutrition (SAM). However, those patients have no medical complication and passed appetite test. Therefore, all included patients fulfill SAM diagnostic criteria but not hospital admission criteria. That means, very low weight for height (below -3z scores of the median WHO growth standards), by visible severe wasting, or by the presence of oedema of both feet and mid-upper arm circumference (MUAC) < 115 mm. this have been explained in the introduction part of the paper. 

Results

Reviewer comment: 11. Results: the section “Bivariate and Multivariate analysis on treatment outcome of SAM and associated factors” shall be re-written as “Treatment outcome of SAM and associated factors”

Authors’ response: amended as per your comment (page 13 line 291)

Reviewer comment: 12. Under the section “Treatment outcome of SAM and associated factors” it remains important to present the length of treatment and related outcome. As it has several advantages for policy and follow up.

Authors’ response: The length of stay for SAM management in OTP program were ranges from 2 months up to 4 months. However, the length of stay was not associated with the treatment outcome in our study (Page 10, line 263-264).

Reviewer comment: 13. “Table 4: Bivariate and Multivariate analysis on treatment outcome of SAM and associated factors” shall be re-written as “Table 4: Treatment outcome of severe acute malnutrition and associated factors in ------ Woreda, Northern Wollo, Ethiopia ---month, ----year to -----month, ----year”

Authors’ response: Amended as per your comment (Page 14 line 310-312)

Reviewer comment: 14. Discussion: This section shall present the main findings obtained by analyzing the data in line with the objective and discuss contextually. “The study was mainly aimed to indicates treatment outcomes of OTP and associated factors with it among children treated from SAM” This is vague and requires clarification. Did you address “Accordingly, the overall prevalence of cured, dead, defaulter, and medical transfer were 65.0, 2.0, 16.0, and 17.0 respectively” in your study adequately

Authors’ response: the aim of the study was to assess treatment outcomes of SAM in OTP and associated factors. As per our objective we have tried to address both objectives. Because regarding the first objective (treatment outcome of SAM) we found overall prevalence of cured, dead, defaulter, and medical transfer were 65.0, 2.0, 16.0, and 17.0 respectively. Regarding the second objective (factor affecting treatment outcome-recovered/not recovered) we found : presence of cough, the presence of diarrhea PO antibiotics, admission category, and the immunization status of a child as predictor of SAM treatment outcome. We have tried to discuss accordingly.

Regarding the first objective: the prevalence of treatment outcome (cured, dead, defaulter, and medical transfer) was found to be 65.0, 2.0, 16.0, and 17.0 respectively

Regarding the second objective: Factors affecting treatment outcome we indicated in the discussion like vaccinated vs unvaccinated children, newly admitted vs re-admitted children page 10 line 343-353.

---

## [Decision Letter · Decision Letter 1]

13 Aug 2020

Treatment outcome of Severe Acute Malnutrition and associated factors among under-five children in outpatient therapeutics unit in Gubalafto Wereda, North Wollo Zone, Ethiopia, 2019

PONE-D-19-29446R1

Dear Dr. Beletew,

We’re pleased to inform you that your manuscript has been judged scientifically suitable for publication and will be formally accepted for publication once it meets all outstanding technical requirements.

Kind regards,

Ricardo Q. Gurgel, PhD

Academic Editor

PLOS ONE

Additional Editor Comments (optional):

Reviewers' comments:

Reviewer's Responses to Questions

**Comments to the Author**

1. If the authors have adequately addressed your comments raised in a previous round of review and you feel that this manuscript is now acceptable for publication, you may indicate that here to bypass the “Comments to the Author” section, enter your conflict of interest statement in the “Confidential to Editor” section, and submit your "Accept" recommendation.

Reviewer #2: (No Response)

2. Is the manuscript technically sound, and do the data support the conclusions?

Reviewer #2: (No Response)

3. Has the statistical analysis been performed appropriately and rigorously? 

Reviewer #2: (No Response)

4. Have the authors made all data underlying the findings in their manuscript fully available?

Reviewer #2: Yes

5. Is the manuscript presented in an intelligible fashion and written in standard English?

Reviewer #2: (No Response)

6. Review Comments to the Author

Reviewer #2: I found most comments addressed. Authors need to check the language before it is submitted for final publication.

7. PLOS authors have the option to publish the peer review history of their article (what does this mean?). If published, this will include your full peer review and any attached files.

Reviewer #2: No

---

## [Editor Report · Acceptance letter]

20 Aug 2020

PONE-D-19-29446R1 

Treatment outcome of Severe Acute Malnutrition and associated factors among under-five children in outpatient therapeutics unit in Gubalafto Wereda, North Wollo Zone, Ethiopia, 2019 

Dear Dr. Abate:

I'm pleased to inform you that your manuscript has been deemed suitable for publication in PLOS ONE. Congratulations! Your manuscript is now with our production department. 

Kind regards, 

on behalf of

Professor Ricardo Q. Gurgel 

Academic Editor

PLOS ONE